# Mechanism-Oriented Analysis of Core–Shell Structured CIP@SiO_2_ Magnetic Abrasives for Precision-Enhanced Magnetorheological Polishing

**DOI:** 10.3390/mi16050495

**Published:** 2025-04-24

**Authors:** Chunyu Li, Shusheng Chen, Zhuoguang Zheng, Yicun Zhu, Bingsan Chen, Yongchao Xu

**Affiliations:** 1Fujian Key Laboratory of Intelligent Machining Technology and Equipment, Fujian University of Technology, Fuzhou 350118, China; 19801546@fjut.edu.cn (C.L.); dhgy134679@163.com (S.C.); 19862091@fjut.edu.cn (Y.X.); 2Castech Inc., Fuzhou 350003, China; zhuyicun@castech.com; 3China National Nuclear Corporation (CNNC), Ningde 355200, China

**Keywords:** CIP@SiO_2_ composite particles, magnetorheological polishing, surface roughness, material removal rate

## Abstract

This study addresses the critical challenge of precise control over active abrasive particles in magnetorheological polishing (MRP) through innovative core–shell particle engineering. A sol–gel synthesized CIP@SiO_2_ magnetic composite abrasive with controlled SiO_2_ encapsulation (20 nm shell thickness) was developed using tetraethyl orthosilicate (TEOS) as the silicon precursor, demonstrating significant advantages in optical-grade fused silica finishing. Systematic polishing experiments reveal that the core–shell architecture achieves a remarkable 20.16% improvement in surface quality (Ra = 1.03 nm) compared to conventional CIP/SiO_2_ mixed abrasives, with notably reduced surface defects despite a modest 8–12% decrease in material removal rate. Through synergistic analysis combining elastic microcontact mechanics modeling and molecular dynamics simulations, we establish that the SiO_2_ shell mediates stress distribution at tool–workpiece interfaces, effectively suppressing deep subsurface damage while maintaining nano-scale material removal efficiency. The time-dependent performance analysis further demonstrates that extended polishing durations with CIP@SiO_2_ composites progressively eliminate mid-spatial frequency errors without introducing new surface artifacts. These findings provide fundamental insights into designed abrasive architectures for precision finishing applications requiring sub-nanometer surface integrity control.

## 1. Introduction

Magnetorheological polishing (MRP) technology is an enlarged use of the magnetorheological effect. It boasts excellent processing accuracy, severe convergence requirements, good surface quality, and minimum subsurface damage. MRP has been applied to polishing optical elements such as sapphire, silicon carbide, and GaN wafers [1,2,3]. The polishing principle involves employing magnetorheological fluid, which changes into a visco-plastic medium under the influence of a magnetic field. This generates a “flexible polishing mode” upon contact with the part’s surface, generating a considerable shear force to accomplish stable material removal, thus ensuring both global and local flatness of the processed surface and reaching the ultimate goal of flexible ultra-precision polishing [4,5]. Magnetorheological polishing fluid, a crucial component of MRP technology, consists of a mixture of carrier liquid, magnetic particles, deionized water, abrasives, and additives. In the absence of a magnetic field, the abrasive particles are suspended and free, behaving like a Newtonian fluid with low viscosity. When a magnetic field is applied, a shearable magnetorheological polishing liquid is generated. The magnetic force chains convey the abrasive particles and workpiece, causing relative sliding to remove surface materials, thus boosting surface accuracy [6,7]. Consequently, the polishing result is greatly influenced by the magnetorheological polishing fluid’s performance.

The main goal of current research on magnetorheological polishing fluids is to improve their characteristics by varying component ratios and adding different materials. In order to solve low processing efficiency and challenging process control difficulties in polishing technology, Xu et al. [8] created a polymer-based magnetorheological elastic polishing composite made of rubber and micro- and nanoparticles. Zhao [9] investigated the microstructure and shear characteristics of the interface between dimethyl silicone oil (DSO) and carbonyl iron particles (CIPs) using molecular dynamics simulations. The findings showed that during the shearing process, there was considerable adhesive strength between the DSO molecular chain and the Fe atoms on the surface of the CIPs. Niranjan M et al. [10] prepared a bidisperse magnetorheological polishing solution composed of micron CS and HS carbonyl iron powder (CIP) and compared the surface roughness obtained using this solution with existing monodisperse magnetorheological polishing solutions, achieving better results. In order to improve the rheological characteristics of magnetorheological polishing fluid, Cao et al. [11] added α-cellulose, which they found increased the fluid’s apparent viscosity and stability. Super-smooth surfaces can be obtained with mixed abrasive, a popular material removal technique in conventional chemical mechanical polishing. D Song et al. [12] studied the effect of different surfactant combinations on the settling stability and viscosity of magnetorheological polishing fluids. The results showed that using a combination of 0.25 mL of Span 80 and 0.25 mL of Tween 60, the settling rate of the magnetorheological solution was 68.5%, and the zero field viscosity was 2.3 mPa · s. Polishing experiments were conducted using configured magnetorheological fluids, and the surface roughness of the workpiece was significantly improved with good results. R Milde et al. [13] prepared a deposition stable magnetic current (MR) polishing slurry based on ferromagnetic fluid, iron particles, Al_2_O_3_, and olivine form clay nanofillers. The findings demonstrated that despite retaining the slurry’s good abrasive qualities, the addition of clay greatly increased its settling stability, lowering its settling rate to a quarter of its initial value. The driving effect of the magnetorheological effect on abrasive particles is yet unknown, despite the fact that it is commonly used in current magnetorheological polishing solutions.

Addressing the uniform distribution of abrasive materials to improve polishing efficiency and accuracy has garnered significant attention and research [14,15]. A key research direction involves preparing composite abrasives with core–shell structures. For example, Zhou C et al. [16] prepared PS@CeO_2_, PS@DND, and PS@CeO_2_/DND composite abrasives using electrostatic attraction between negatively charged PS colloid and cerium cations or positively charged diamond particles; after polishing using the ternary abrasive PS@CeO2/DND, the sapphire surface could be smoothed to sub-nanometer roughness (Ra = 0.52 nm), and the MRR of the ternary composite abrasive reached 1.4–1.7 μmh^−1^. Wang L et al. [17] introduced a simple precipitation method to uniformly deposit CeO_2_ on SiO_2_, prepared SiO_2_@CeO_2_ core–shell particles, and incorporated them into chemical mechanical polishing slurry. The material removal rate (MRR) increased from 32.27 Å/min to 294.02 Å/min. Chen Z et al. [18] synthesized TiO_2_-CI functional composite particles using a sol–gel method and applied them in the magnetorheological polishing of silicon wafers and fused silica glass with photocatalytic assistance. The results showed that TiO_2_-CI composite particles formed a flexible polishing pad under a magnetic field, absorbed the abrasive, and mechanically removed workpiece material, while photocatalytic reactions generated strong hydroxyl radicals (·OH) to oxidize surface material. With the assistance of the light field, the MRR increased by 249% to 13.4 mg/h, and the surface roughness decreased by 69.32% to 1.05 nm. Q Zhai et al. [19] successfully prepared Fe_3_O_4_/SiO_2_ (core/shell) composite abrasives with different core diameters and shell thicknesses by adjusting the amount of TEOS in the hydrolysis reaction and studied the effects of the core diameter and shell thickness of the composite abrasives on the ultrasonic-assisted magnetorheological polishing (UAMP) performance of sapphire chips. The results showed that when using a composite abrasive magnetorheological slurry with a shell thickness of 15 nm, the formation rate of softer chemical products on the sapphire surface and their mechanical removal rate almost reached equilibrium, with Ra and MRR reaching the optimal values of 0.276 nm and 2.068 µ m/h, respectively. Lee J W et al. [20] coated CIPs with polymethyl methacrylate for improved magnetorheological polishing and corrosion protection of CIPs. M Sedlacik et al. [21] coated a thin layer of (3-aminopropyl) triethoxysilane (3APTS) with a density of approximately 50 functional groups per nano2 onto soft magnetic carbon iron (CI) particles. The results showed that the coating did not affect the morphology and magnetism of the particles, and their resistance to thermal oxidation and chemical degradation was significantly improved. Compared with the uncoated CI particle-based suspension, the compatibility between CI particles coated with 3APTS and silicone oil was improved. Recently, various materials such as graphene oxide (GO), aluminum oxide (Al_2_O_3_), and silicon dioxide (SiO_2_) have been used to modify the surface of CIPs [22,23,24]. SiO_2_, as a good wave-transparent medium, can prevent magnetic particles’ oxidation, improve their chemical stability, and promote dispersion [25].

The research has found that core–shell structure abrasives are often used to increase the reactive phase or to form soft and hard composite abrasives to enhance surface quality. The goal of composite particles, which integrate magnetic and abrasive particles through a core–shell structure, is to increase the efficiency and uniformity of abrasive particle driving in magnetorheological polishing fluid. In order to create a uniform SiO_2_ coating layer on magnetic particles and create CIP@ SiO_2_ magnetic composite particles, which are characterized and validated, this paper suggests using the hydrolysis and condensation reaction of tetraethyl orthosilicate (TEOS) in an alkaline environment. Through magnetorheological polishing studies, the impacts of particles with a core–shell structure versus mixed abrasive particles on polishing surface quality were compared.

## 2. Materials and Methods

### 2.1. Chemicals and Materials

In the preparation of CIP@SiO_2_ magnetic composite particles, CIP (Jiangsu Tianyi Ultrafine Metal Powder Co., Ltd., Jiangsu, China) is used as the core material. Tetraethyl orthosilicate (TEOS, (C_2_H_5_O)_4_Si, analytical grade, Tianjin Damao Chemical Reagent Factory, Tianjin, China) provides the SiO_2_ shell material through hydrolysis. Anhydrous ethanol (C_2_H_5_OH, analytical grade, Jiangsu Qiangsheng Functional Chemicals Co., Ltd., Jiangsu, China) and ammonia solution (NH_3_·H_2_O, 25%, Xilong Scientific Co., Ltd., Shantou, China) are used in the process. For comparison experiments, nanosilica (Beijing Shenghe Haoyuan Technology Co., Ltd., Beijing, China) is used as the abrasive. Deionized (DI) water is employed throughout the polishing experiments.

### 2.2. Preparation Principle and Process of Magnetic Composite Particles

In this experiment, the sol–gel method is employed to synthesize CIP@SiO_2_ magnetic particles. The reaction principle is to use TEOS to hydrolyze in an alkaline environment to produce silanol (Si-OH). The surface of CIP enhances the surface hydroxyl groups (Fe-OH) in an alkaline environment. Then, Si-OH and Fe-OH undergo an interfacial condensation reaction to form Fe-O-Si covalent bonds, and Si-OH condenses itself to form a SiO_2_ network, and finally forms a SiO_2_ shell that is adsorbed on CIPs [26,27,28]. Figure 1 shows a schematic diagram of CIP@SiO_2_ synthesis.

In order to ensure the purity of CIP, a specific quantity of iron powder was weighed and then cleaned with 100% ethanol to get rid of any contaminants that may have mixed in. After completely dispersing it with ultrasonic agitation, it was put in a vacuum drying oven to finish drying. Then, 9.8 g of powdered carbonyl iron was weighed and put in a beaker. The beaker was filled with 180 mL of 100% ethanol; then, ultrasonography was used to disperse it for 20 min. Then, 12.48 g of ethyl orthosicate, 40 mL of deionized water, and 6 mL of ammonia were added, in that order. At a reaction temperature of 35 °C and a rotational speed of 500 r/min, mechanical stirring was performed for 12 h. Following the reaction, the iron powder was dried for 24 h at 60 °C after being accelerated in its sedimentation by a magnet. The specific procedure is depicted in Figure 2.

### 2.3. Property Measurement and Characterization of CIP@SiO_2_ Magnetic Composite Particles

X-ray examination was carried out to confirm whether structural alterations took place throughout the coating process. As shown in Figure 3, three identical diffraction peaks were observed at 2θ = 44.7°, 65.03°, and 82.36° for both CIPs and CIP@SiO_2_ composite particles, respectively. According to (JCPDS No. 87-0721) [29], these three diffraction peaks correspond to the (110), (200), and (211) crystal planes of the body-centered cubic structure α-Fe. As shown by the dashed box in Figure 3, a single diffraction peak was observed in the CIP@SiO_2_ composite particle pattern at 2θ = 12° to 18°, which is a characteristic peak of amorphous SiO_2_. The diffraction peak is broad and low. The coated CIP@SiO_2_ composite particle has good purity and has not altered its original structure, as seen by the absence of any additional impurity peaks in the CIP@SiO_2_ composite particle pattern when compared to the CIP pattern.

The elemental composition of CIP@SiO_2_ magnetic composite particles and pure CIPs was characterized by scanning electron microscopy (FESEM). As shown in Figure 4, (a) and (b) show the EDS spectra of CIP@SiO_2_ magnetic composite particles and pure CIPs, respectively. Compared with CIPs, in addition to the Fe, C, and O elemental peaks inherent in CIPs, CIP@SiO_2_ magnetic composite particles have an additional peak for the Si element at 1.740 keV, with an atomic percentage of 3.34%.

The element distribution of CIP@SiO_2_ composite magnetic particles and CIPs is shown in Figure 5. Compared with CIPs, CIP@SiO_2_ contains more Si elements in addition to Fe, C, and O, which are inherent in CIP. The Si element comes from the surface-coated silica shell. It can be seen from Figure 5b that the Si element is distributed on the surface of the CIP microspheres, indicating that the silicon element is coated on the surface of CIP.

CIP@SiO_2_ magnetic composite particles and CIPs were observed using TEM. As shown in Figure 6, (a–c) are images of the same CIP@SiO_2_ magnetic composite particle at different magnifications, and (d–f) are images of the same CIP at different magnifications. Compared with CIsP, it can be clearly seen that on the CIP surface of the TEM image of the CIP@SiO_2_ magnetic composite particle, a layer of translucent material is attached, with an obvious core–shell structure, and the outer shell is uniformly distributed. From this, it can be concluded that a newly generated material is uniformly coated on the outside of the CIP and has a diameter thickness of approximately 20 nm.

The CIP@SiO_2_ magnetic composite particles and CIPs were tested using a vibrating sample magnetometer (VSM). Figure 7 shows the hysteresis loops of the two types of particles. The hysteresis loops are similar in shape, indicating that both are soft magnetic materials. When the particles reach magnetic saturation, the externally applied magnetic field strength is approximately equal. However, the saturation magnetization of CIP is 210.1 emu/g, while that of the CIP@SiO_2_ composite particles decreases to 202.2 emu/g due to the presence of the silica shell. Therefore, during surface finishing, the CIP@SiO_2_ composite particles exhibit lower magnetic saturation and are more prone to concentrating in the finishing area.

## 3. Experiment Design for the MRP

### 3.1. Principle and Device of Magnetorheological Polishing

The experimental object of magnetorheological polishing in this paper is a fused silica sample with high purity and an amorphous structure with a Mohs hardness of around 6. It has good physical and chemical properties, including low density (2.20 g/cm³), low thermal conductivity (0.55 × 10/K), good light transmission, stability, and corrosion resistance, making it widely used in the aerospace industry, microelectronic chips, and national defense and military fields [30,31]. The diameter of the sample is 2 cm and the thickness is 4 mm. Compared with the sample after polishing, the surface of the sample before polishing is rough, and the smoothness is poor, with obvious defects such as pits and bumps visible on the surface microstructure, as illustrated in Figure 8. The DMG five-axis machining center is utilized to design the polishing device for the processing platform, as shown in Figure 9. For the polishing device and its schematic diagram used in this experiment, the processing platform is shown in Figure 9c. BT40 tool holder is used to clamp the carrier plate, and the polished sample is bonded to the carrier plate through paraffin. At the same time, to improve polishing efficiency, the polished sample and the carrier plate are connected eccentrically, as shown in Figure 9d. Right below it is a slurry tank carrying magnetorheological fluid, and the bottom of the slurry tank is a magnetic field generator. As shown in Figure 9a, a permanent magnet NdFeB branded N30 is used, arranged at intervals and placed inside the polishing pool.

### 3.2. Parameters Design for the MRP Process

MRP experiments were carried out by using CIP@SiO_2_ composite abrasives and CIP/SiO_2_ mixed abrasives; the particle diameter of the CIPs was 300 nm, and the particle diameter of free SiO_2_ was 20 nm with Mohs hardness of around 7. The differences in surface roughness and material removal rate at different polishing times, polishing speeds, and polishing gaps were compared, and each experiment was repeated three times to eliminate random errors. Specific polishing parameter settings are shown in Table 1. According to Figure 6, TEM images of CIP@SiO_2_ composite particles and CIPs, it can be seen that when the particle size of CIPs is 300 nm, the thickness of the prepared CIP@SiO_2_ shell is about 20 nm. For the preparation principle and process of magnetic composite particles, this paper assumes that CIP@SiO_2_ composite abrasive and CIP/SiO_2_ free abrasive are ideal balls. According to the calculation formula of the ball volume, it can be obtained that the SiO_2_ shell in 35 vol% CIP@SiO_2_ accounts for 11 vol% of the volume of CIP@SiO_2_ polishing fluid. Therefore, in this paper, the concentrations of CIPs and free SiO_2_ in the 35 vol% CIP/SiO_2_ free abrasive are selected to be 24 vol% and 11 vol%, respectively.

### 3.3. Measurement and Calculation of Surface Roughness and Material Removal Rate

After each experiment, the polished sample was placed at the bottom of the beaker and absolute ethanol was poured over it. It was placed in an ultrasonic cleaner for ultrasonic cleaning for 8–10 min. Then, it was rinsed in flowing deionized water for 2–3 min and placed in a dryer to dry. The polished sample’s surface roughness was measured, its surface topography was analyzed, and measurements were taken at five separate locations. To find the overall surface roughness of the polished sample, the average value of these measurements was computed. Figure 10 displays the point picking positions’ schematic diagram.

An electronic analytical balance (GE0505, Shanghai YoKe Instrument, Shanghai, China) with accuracy of 0.01 mg was used to measure the mass of fused silica three times, and the material removal rate was calculated according to Equation (1):(1)$$MRR=\frac{(m_{1}-m_{2})\times 10^{4}}{\rho At}$$
where *MRR* is the corresponding material removal rate (μm/min), *m*_1_ and *m*_2_ denote the weight of the fused silica sample before and after MRP (g), *ρ* denotes the fused silica density (2.2 g/cm^3^), *t* denotes the polishing time (min), and *A* is the surface area for the polished fused silica (3.14 cm^2^).

## 4. Results and Discussion

### 4.1. Effect of Polishing Time on Polishing Effect Under Different Abrasives

One significant element influencing polishing quality is polishing time. This experimental group chose CIP@SiO_2_ composite abrasive and CIP/SiO_2_ mixed abrasive solutions, both with a concentration of 35 vol%, for comparative polishing trials in order to investigate the impact of various abrasives on polishing effects. The surface roughness was assessed every half hour, the polishing period was 120 min, the polishing gap was 0.6 mm, and the polishing speed was 600 r/min.

Figure 11a,b display the change curves for surface roughness and MRR for polishing fused silica using CIP@SiO_2_ composite abrasive and CIP/SiO_2_ mixed abrasive, respectively. The material removal rates with CIP@SiO_2_ composite abrasive and CIP/SiO_2_ mixed abrasive were 0.328 μm/min and 0.342 μm/min, respectively, as seen in Figure 11a. This represents a 4.09% reduction in comparison to CIP/SiO_2_ mixed abrasive. The fused silica’s initial surface roughness was around 8 nm, as seen in Figure 11b. The surface roughness employing CIP@SiO_2_ composite abrasive and CIP/SiO_2_ mixed abrasive quickly dropped to 2.46 nm and 2.14 nm, respectively, after polishing for 30 min. The surface roughness improvement efficiencies were 69.19% and 73.34%, respectively, suggesting that at this level, a CIP/SiO_2_ mixed abrasive can produce improved surface quality more rapidly. However, the surface roughness curve for the CIP/SiO_2_ mixed abrasive progressively smoothed out as the polishing time increased. The two abrasives’ respective surface roughness values after 120 min of polishing were 1.45 and 1.2 nm. The CIP@SiO_2_ composite abrasive polishing resulted in a 17.24% improvement in the final surface roughness. This is primarily due to the fact that free SiO_2_ abrasives are more mobile in the polishing slurry and have a higher hardness than amorphous SiO_2_ attached on CIP surfaces. This allows them to rapidly touch the workpiece surface and effectively remove material to improve surface quality more quickly. A more stable material removal method that prevents over-cutting or surface damage is offered by the softer amorphous SiO_2_ abrasives bonded to CIP surfaces. The softer composite abrasives can sustain more steady polishing pressure, which will ultimately result in higher final surface quality, whereas free abrasives may create surface inhomogeneity due to excessive cutting during lengthy polishing.

The diagonal surface profile curves and surface topography of fused silica polished by various abrasives at 30 and 120 min are displayed in Figure 12. Unremoved peaks and scratches from the previous process were present on both surfaces after 30 min of polishing; these were more removed after 120 min of polishing, but the sample’s surface was more damaged when polishing with CIP/SiO_2_ mixed abrasives. More and longer scratches were discovered on the surface when compared to CIP@SiO_2_ composite abrasive. The diagonal surface profile curve polished with CIP/SiO_2_ mixed abrasive had a wave trough near −10 nm and a wave peak near 10 nm after 30 min of polishing, and the diagonal surface profile curve polished with CIP@SiO_2_ composite abrasive had three wave troughs near −10 nm and three wave peaks near 10 nm. These results show that CIP/SiO_2_ mixed abrasive has a high rate of material removal as well as a high rate of surface defect and peak removal. Following 120 min of polishing, the CIP@SiO_2_ composite abrasive and CIP/SiO_2_ mixed abrasive’s peak–trough values (PV values) of their diagonal surface profile curves dropped from 20.83 nm and 18.82 nm to 8.83 nm and 11.88 nm, respectively. After polishing for 120 min, the diagonal surface profile curves of the two were smoother and the surface quality was significantly enhanced, as seen by the 25.67% reduction when compared to CIP/SiO_2_ mixed abrasive. Of these, polishing the diagonal surface profile curve with CIP@SiO_2_ composite abrasive had less variation, resulting in higher surface quality.

### 4.2. Effect of Polishing Speed on Polishing Effect Under Different Abrasives

The contact frequencies between the abrasives and the sample vary because of the variable relative linear velocities between the abrasives and the sample surface at various polishing speeds. This variance has a major effect on the sample’s surface quality and rate of material removal. The impact of CIP@SiO_2_ composite abrasive and CIP/SiO_2_ mixed abrasive MRP fluids on the polishing effect at varying polishing speeds was examined in this series of tests. The polishing spacing was 0.6 mm, the polishing time was 120 min, and the CIP@SiO_2_ and CIP/SiO_2_ abrasive concentrations were both 35 vol%. The fused silica sample’s initial surface roughness was about 8 nm, and the polishing speeds were 600 r/min, 800 r/min, and 1000 r/min, respectively.

Polished fused silica’s surface roughness values and material removal rates utilizing CIP@SiO_2_ composite abrasive and CIP/SiO_2_ mixed abrasive at various rotating speeds are displayed in Figure 13. With increasing rotational speed, the surface roughness values when polished with CIP@SiO_2_ composite abrasive and CIP/SiO_2_ mixed abrasive first reduced and then increased. Compared to CIP/SiO_2_ mixed abrasive, the surface roughness values for CIP@SiO_2_ composite abrasive were lower. At a rotational speed of 800 r/min, both attained their lowest levels, 1.03 nm and 1.29 nm, respectively. Surface roughness was reduced by 20.16% while polishing with CIP@SiO_2_ composite abrasive as opposed to CIP/SiO_2_ mixed abrasive. As the polishing speed increased, the material removal efficiency first rose and subsequently fell. The maximum removal rates at 800 r/min were 0.391 μm/min for the CIP/SiO_2_ mixed abrasive and 0.373 μm/min for the CIP@SiO_2_ composite abrasive. The material removal effectiveness of the CIP@SiO_2_ composite abrasive was 4.6% lower than that of the CIP/SiO_2_ mixed abrasive. The material removal efficiency decreased by 17.43% and 18.93% to 0.308 μm/min and 0.317 μm/min, respectively, when the rotational speed approached 1000 r/min. The primary cause of this drop was an excessively high polishing rotational speed, which caused the magnetic chain to fracture in reverse under the impact of increased centrifugal force.

The diagonal surface profile curves and surface topography of fused silica polished with CIP@SiO_2_ composite abrasive and CIP/SiO_2_ mixed abrasive at varying rotational speeds are displayed in Figure 14. Compared to utilizing CIP/SiO_2_ mixed abrasive, polishing with CIP@SiO_2_ composite abrasive produces fewer and shallower scratches at the same rotational speed. The amount of material removed per unit of time is minimal when the polishing speed is 600 r/min because the polishing abrasive can only come into contact with the workpiece a limited number of times. For CIP@SiO_2_ composite abrasive and CIP/SiO_2_ mixed abrasive, the diagonal surface profile curves have PV values of 7.54 nm and 12.01 nm, respectively, and they exhibit significant fluctuations. The PV values of the diagonal profile curves for fused silica polished with CIP@SiO_2_ composite abrasive and CIP/SiO_2_ mixed abrasive decreased to 5.56 nm and 8.67 nm, respectively, when the polishing speed was increased to 800 r/min. The PV value of the CIP@SiO_2_ composite abrasive was 35.87% lower at this speed than that of the CIP/SiO_2_ mixed abrasive, suggesting that the CIP@SiO_2_ composite abrasive can produce a higher-quality surface. With scratches from earlier procedures still present and with there being less flatness, the substantial drop in material removal efficiency caused by the polishing speed increase to 1000 r/min resulted in an increase in surface flaws. The diagonal surface profile curves showed an increase in PV values to 15.03 nm and 19.31 nm, respectively.

### 4.3. Effect of Polishing Gap on Polishing Effect Under Different Abrasives

The thickness of the polishing pad created by the polishing liquid and the polishing force exerted by the abrasive particles on the polished sample vary in different polishing gaps, which has a significant effect on the rate of material removal and surface quality. This experimental group examined the effects of using CIP@SiO_2_ composite abrasive and CIP/SiO_2_ mixed abrasives on polishing under three different gaps: 0.3 mm, 0.6 mm, and 0.9 mm. This is because the surface of the CIP@SiO_2_ composite abrasive is coated with a soft SiO_2_ shell and has a lower hardness than the SiO_2_ abrasive grains. The fused silica sample’s initial surface roughness was about 8 nm, the polishing speed was 600 r/min, and the CIP@SiO_2_ and CIP/SiO_2_ concentrations were both at 35 vol%. The polishing process lasted 120 min.

The surface roughness and material removal rate of polished fused silica with CIP@SiO_2_ composite abrasive and CIP/SiO_2_ mixed abrasive under various spacing are displayed in Figure 15. As the polishing gap widens, the two’s surface roughness initially decreases and then grows. When the spacing is 0.6 mm, the minimum surface roughness of the CIP@SiO_2_ composite abrasive polishing is 1.23 nm, which is 14.58% rougher than the minimum surface roughness of the CIP/SiO_2_ mixed abrasive polishing, which is 1.44 nm. The material removal rates of CIP@SiO_2_ composite abrasive and CIP/SiO_2_ mixed abrasive are 0.412 μm/min and 0.387 μm/min, respectively, when the polishing gap is 0.3 mm. Compared to the CIP/SiO_2_ mixed abrasive, the CIP@SiO_2_ composite abrasive is 6.07% less effective. The depth of abrasive grains forced into the sample’s surface decreases and the sample is cut less as the polishing gap widens. The rate of material removal gradually declines. The material removal rate decreases to 0.323 μm/min when the polishing gap is 0.9 mm, and the CIP@SiO_2_ composite abrasive is 4.95% less effective than the CIP/SiO_2_ mixed abrasive at 0.307 μm/min.

Figure 16 shows the surface topography and diagonal surface profile curve of fused silica polished with CIP@SiO_2_ composite abrasive and CIP/SiO_2_ mixed abrasive under different polishing gaps. When the polishing gap is 0.3 mm, the polishing abrasive grains are closely packed and exert a large polishing force on the sample surface. The abrasive grains enter a plowing or cutting state, resulting in scratches on both surfaces and causing damage to the sample. The diagonal surface profile curve for the sample polished with CIP/SiO_2_ mixed abrasive has a PV value of 17.59 nm. When polished with CIP@SiO_2_ composite abrasive, the profile curve shows a trough of about −7 nm, and the PV value is 12.79 nm, which is 29.05% lower than that of the CIP/SiO_2_ mixed abrasive. This indicates that the surface scratches from polishing with CIP@SiO_2_ composite abrasive are shallower and less frequent. This is mainly because the SiO_2_ shell of the CIP@SiO_2_ composite abrasive has lower hardness than SiO_2_ alone, resulting in a weaker plowing effect and less surface damage. When the polishing gap is increased to 0.6 mm, surface scratches are reduced with both CIP@SiO_2_ composite abrasive and CIP/SiO_2_ mixed abrasive, with the lowest diagonal surface profile curve PV values being 8.79 nm and 12.07 nm, respectively. Compared to CIP/SiO_2_ mixed abrasive, the reduction with CIP@SiO_2_ composite abrasive is 27.17%. As the polishing gap increases to 0.9 mm, the material removal rate decreases, leading to an increase in surface defects on both sides. The PV values of the diagonal profile curve rise to 12.02 nm and 13.99 nm, respectively, with increased fluctuations in the curve and decreased surface quality.

### 4.4. Mechanism Analysis for the Polishing

The MRP process of fused silica mainly involves the mechanical interaction between abrasive grains and polished samples. Building on this, and using elastic–plastic micromechanics combined with LAMMPS molecular dynamics simulations [32], the contact behavior between abrasives and the workpiece is studied, revealing the mechanical effects during the polishing process. Figure 17 is a schematic diagram of the micro-contact between particles and fused silica samples during polishing. According to the elastic micro-contact mechanical model proposed by Chen et al. [33]:(2)$$\delta_{W}=d_{1}-\delta \;\;\;\;\;$$(3)$$\delta =(\frac{9F^{2}}{8DE_{SW}^{2}}{)}^{1/3}$$(4)$$\frac{1}{E_{SW}}=\frac{1-\nu_{s}^{2}}{E_{S}}+\frac{1-v_{w}^{2}}{E_{W}}$$

*δ_w_* denotes the indentation depth at which the particles enter the wafer, and *d*_1_ denotes the depth at which the particles enter the wafer regardless of the deformation of the particles. *F* denotes the polishing pressure, *D* is the particle diameter, and *E_sw_* is the Young’s modulus of the particle and the wafer. *E_S_* and *ν_S_* are the Young’s modulus and Poisson’s ratio of the abrasive particles, respectively. It can be seen from Equations (3) and (4) that under the action of polishing pressure *F*, the lower the elastic modulus *E_S_* of the abrasive grains, the greater the deformation *δ* occurring during the polishing process, so that the indentation depth *δ_w_* of the abrasive grains pressed into the fused silica sample decreases [33]. The indentation depth during the polishing process has a great influence on the final mechanical damage and surface roughness of the fused silica sample. The CIP@SiO_2_ composite abrasive prepared in this paper has a soft SiO_2_ shell, and its overall elastic modulus is lower than that of pure SiO_2_ abrasive grains. Therefore, the CIP@SiO_2_ composite abrasive reduces the depth of the mark after polishing, effectively avoiding deep scratches and serious damage.

To further verify the difference between SiO_2_ shell and solid SiO_2_ abrasive grains in removing fused silica material, molecular dynamics simulations were conducted on the process of processing fused silica with single abrasive grains using LAMMPS. Figure 18 shows a molecular dynamics simulation model for processing fused silica with SiO_2_ abrasive grains and SiO_2_ shell abrasive grains. The dimensions of the fused silica matrix are approximately 216 Å × 157 Å × 104 Å in the x, y, and z directions, respectively. The diameters of the SiO_2_ shell and solid SiO_2_ abrasive grains are both 40 Å. The potential function used is the three-body Tersoff potential, and the applied pressure is 150 nN for both cases.

As shown in Figure 19a,b, the surface topography of fused silica at a processing distance of 100 Å between SiO_2_ abrasive grains and the SiO_2_ shell is examined. The width of the scratch surface accumulation of SiO_2_ abrasive grains is 50 Å, and the width of the groove is 35 Å. The width of the scratched surface accumulation of the SiO_2_ shell is 45 Å, and the width of the groove is 29 Å. The width of the groove and the width of the atomic accumulation are both smaller than those observed with SiO_2_ abrasive grains. As shown in Figure 19c,e, the atomic accumulation diagrams of the processed surface using SiO_2_ abrasive grains and the SiO_2_ shell, respectively, clearly indicate that there is significant surface atomic accumulation when using the SiO_2_ abrasive grains. Figure 19d,f show the atomic accumulation morphology of SiO_2_ abrasive grains and the SiO_2_ shell processed fused silica, respectively, with the main processing area sliced at 10 Å in the Z direction. the cross-sectional position is shown in the dashed box in Figure 19a,b. It can be seen that when the processing distance is 100 Å, the shape of the processing grooves with SiO_2_ abrasive grains tends to be more stable than that with the SiO_2_ shell, and the SiO_2_ shell exhibits more elastic recovery, indicating that SiO_2_ abrasive grains have a higher material removal rate compared to SiO_2_ shell abrasive grains.

As shown in Figure 20a,b, the sub-surface atomic displacements of SiO_2_ abrasive grains and SiO_2_ shell processed fused silica, respectively, are illustrated. It can be observed that the atomic displacements during fused silica processing are not like the obvious linear slips of atoms observed during crystal processing, but rather irregular movements. As shown in Figure 20c,d, the displacements of subsurface atoms in the vertical direction of the main processing area processed by SiO_2_ abrasive grains and SiO_2_ shell processed fused silica are, respectively, 10 Å slicing in the Z direction. It can be seen that in the middle processing grooves, the subsurface displacement of abrasive grains is greater, and the atomic displacement is significantly larger where the workpiece is in contact with abrasive grains. Subsurface atomic displacement is defined as subsurface damage in the fused silica. It can be seen that the final damage thickness of SiO_2_ abrasive grains to fused silica is about 16 Å, and the final damage thickness of SiO_2_ shell to fused silica is about 12 Å. Therefore, the SiO_2_ shell helps to reduce the subsurface damage caused to fused silica.

## 5. Conclusions

The sol–gel method was used in this investigation to create CIP@SiO_2_ magnetic composite particles. The original structure of the CIP was unaffected by the coating procedure, according to XRD measurements. The surface of the CIPs was coated with a SiO_2_ shell that measured around 20 nm, as confirmed by mapping diagrams, EDS spectroscopy, and TEM pictures. The CIP@SiO_2_ composite abrasive possesses stronger paramagnetic characteristics, according to VES tests.

Comparative studies were conducted on the magnetorheological polishing of fused silica samples using CIP@SiO_2_ free and CIP@SiO_2_ composite abrasives. The CIP@SiO_2_ composite abrasive and the CIP/SiO_2_ free abrasive achieved the best surface roughness of 1.03 nm and 1.29 nm, respectively. There was a 20.16% rise in the CIP@SiO_2_ composite abrasive. At this point, the CIP@SiO_2_ composite abrasive dropped by 4.6%, and the two material removal rates were 0.373 μm/min and 0.391 μm/min, respectively. This demonstrates that CIP@SiO_2_ composite abrasive may be applied more uniformly than CIP/SiO_2_ free abrasive in the finishing area, enhancing the processed sample’s flatness.

Mechanism analysis was performed from the viewpoints of molecular dynamics and elastic micro-contact mechanics. According to the results of the elastic micro-contact mechanical study, the soft SiO_2_ shell abrasive grains’ lower elastic modulus causes a greater amount of deformation (δ) during the polishing process when the same polishing force is applied. Molecular dynamics simulation results indicate that using SiO_2_ abrasive grains with obvious surface atomic accumulation has a higher material removal rate than SiO_2_ shell abrasive grains. This is because the indentation depth *δ_w_* of the abrasive grains pressed into the fused silica sample from CIP@SiO_2_ composite abrasive grains decreases, improving the surface quality. In line with the experimental findings, the SiO_2_ shell contributes to the enhancement of fused silica’s surface quality. This provides a polishing reference for magnetorheological fluids with regard to ultra-precision machining.

## Figures and Tables

**Figure 1 micromachines-16-00495-f001:**
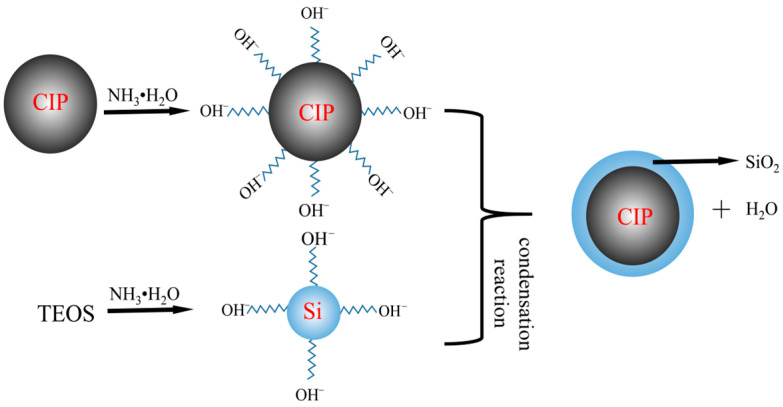
CIP@SiO_2_ synthesis diagram.

**Figure 2 micromachines-16-00495-f002:**
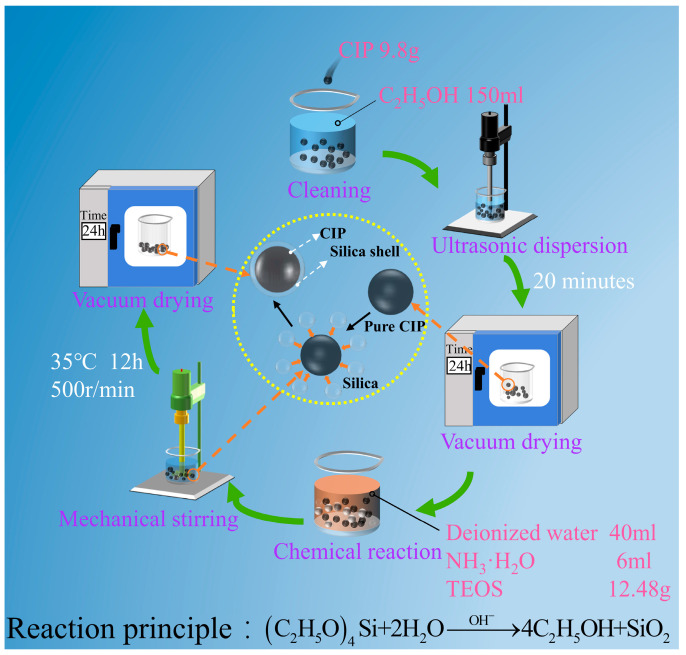
CIP@SiO_2_ preparation process diagram of magnetic composite particles.

**Figure 3 micromachines-16-00495-f003:**
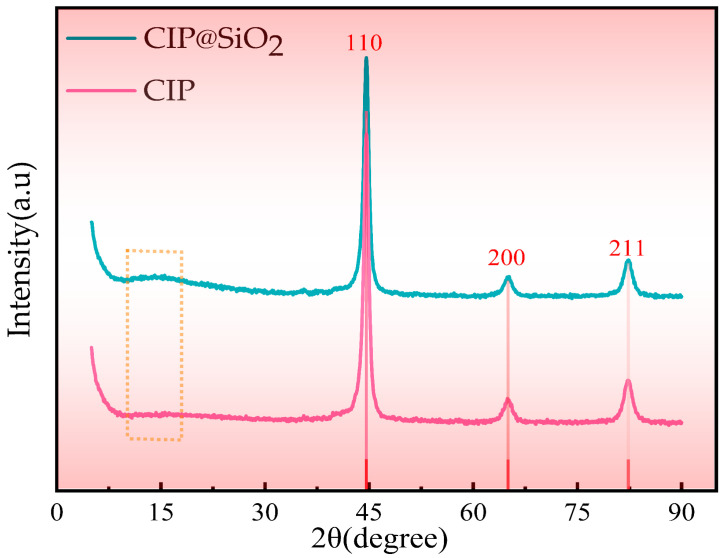
XRD images of CIP@SiO_2_ composite particles and CIPs.

**Figure 4 micromachines-16-00495-f004:**
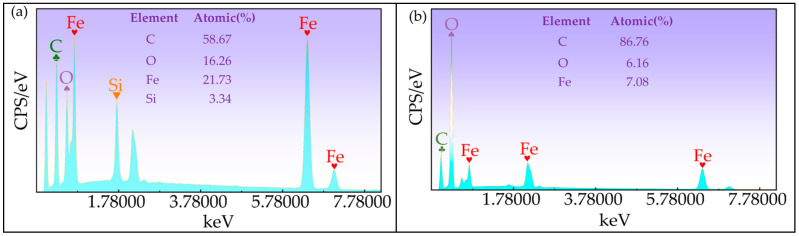
EDS spectral images of CIP@SiO_2_ composite particles (**a**) and CIPs (**b**).

**Figure 5 micromachines-16-00495-f005:**
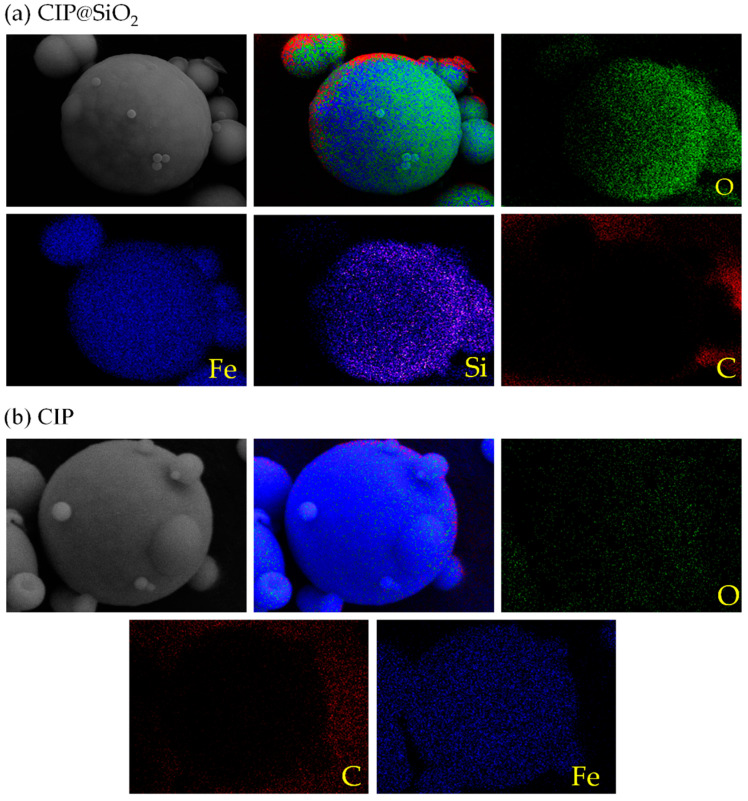
Element distribution images of CIP@ SiO_2_ composite particles (**a**) and CIPs (**b**).

**Figure 6 micromachines-16-00495-f006:**
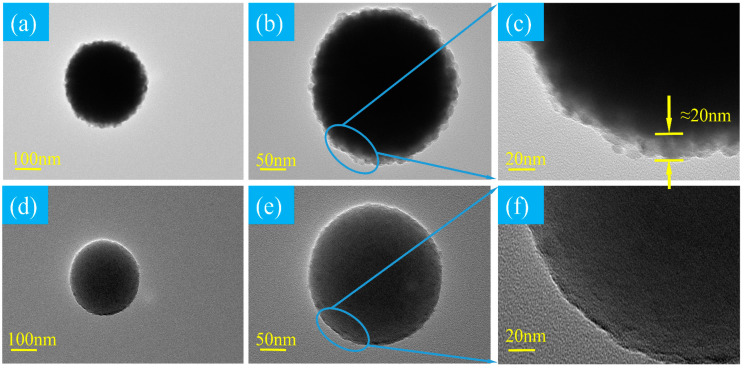
TEM images of CIP@SiO_2_ composite particles (**a**–**c**) and CIPs (**d**–**f**).

**Figure 7 micromachines-16-00495-f007:**
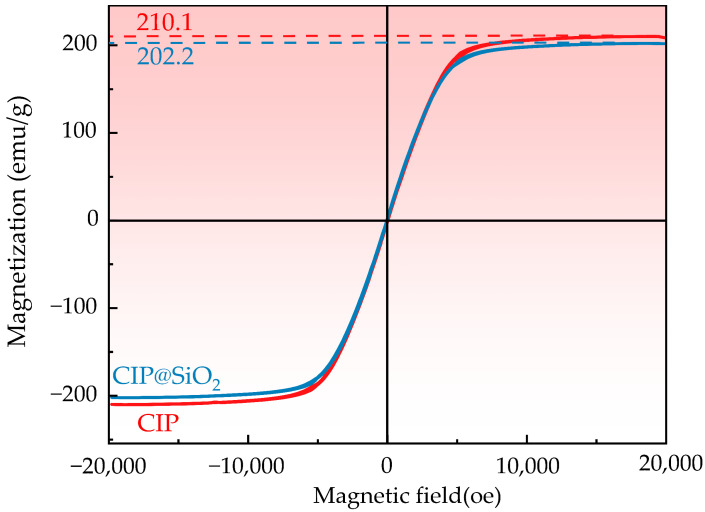
VSM test image of CIP@SiO_2_ composite particles and CIPs.

**Figure 8 micromachines-16-00495-f008:**
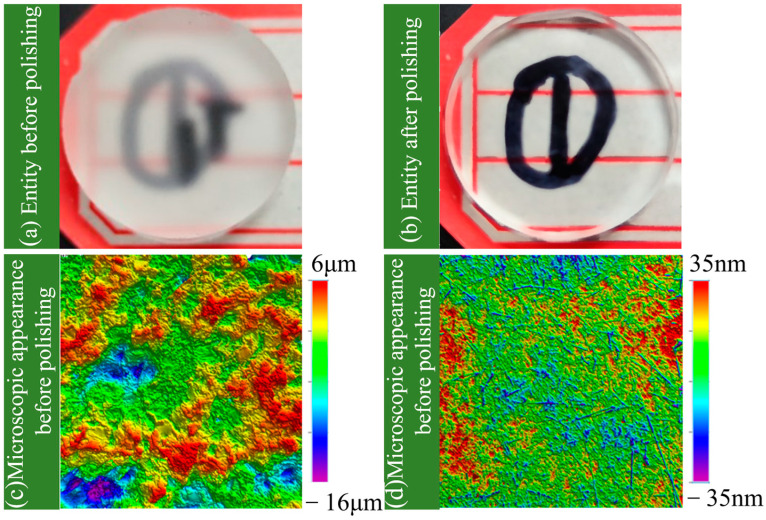
Sample polished with 35% volume CIP@SiO_2_ comparison chart of magnetorheological fluid before and after polishing at a speed of 800 r/min for 2 h in a 0.6 mm polishing gap.

**Figure 9 micromachines-16-00495-f009:**
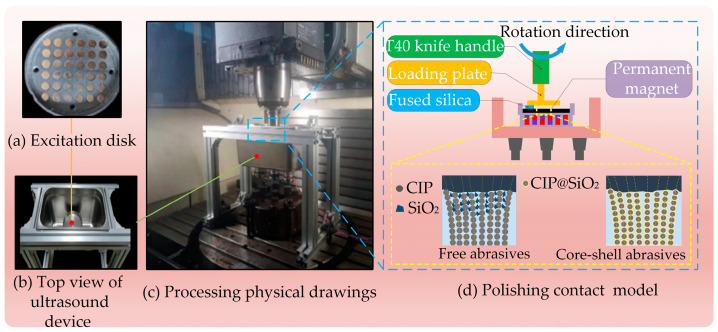
Principle diagram of processing device.

**Figure 10 micromachines-16-00495-f010:**
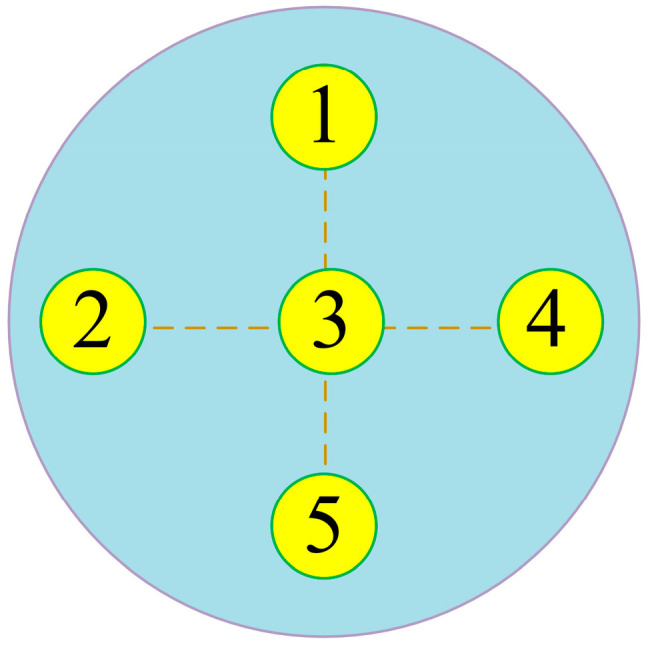
Schematic diagram of surface roughness measurement point location.

**Figure 11 micromachines-16-00495-f011:**
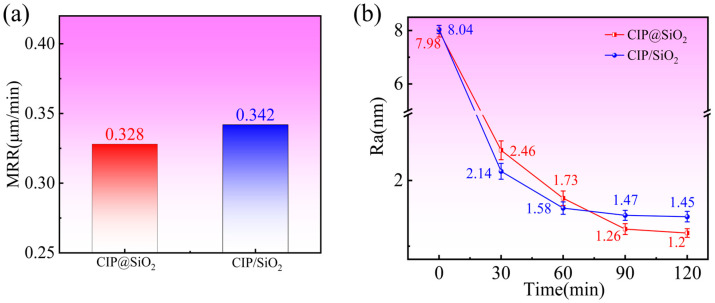
Material removal rate (**a**) and surface roughness (**b**) of fused quartz polished with different abrasives.

**Figure 12 micromachines-16-00495-f012:**
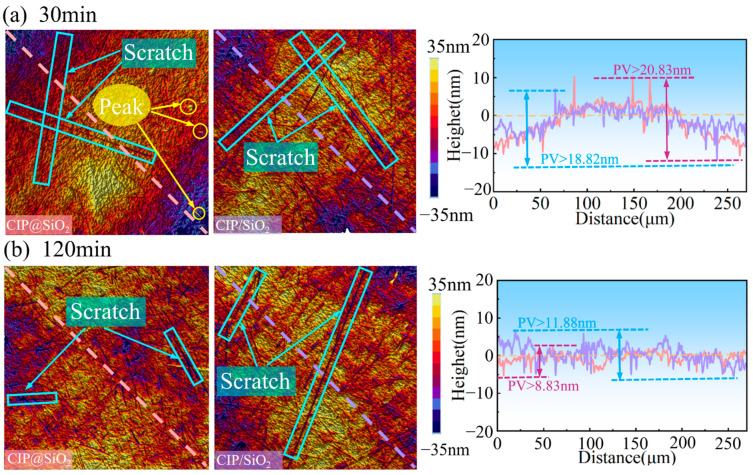
Surface topography and diagonal surface profile curves of fused silica polished with different abrasives and times.

**Figure 13 micromachines-16-00495-f013:**
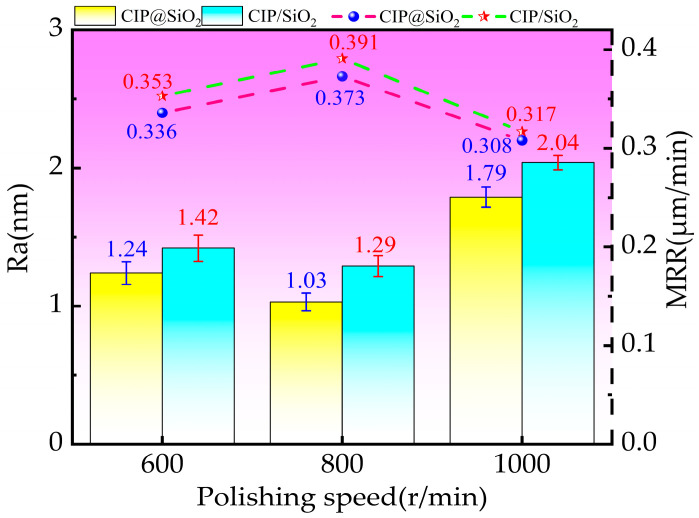
Surface roughness and material removal rate of polished fused silica with different abrasive grains at different rotational speeds.

**Figure 14 micromachines-16-00495-f014:**
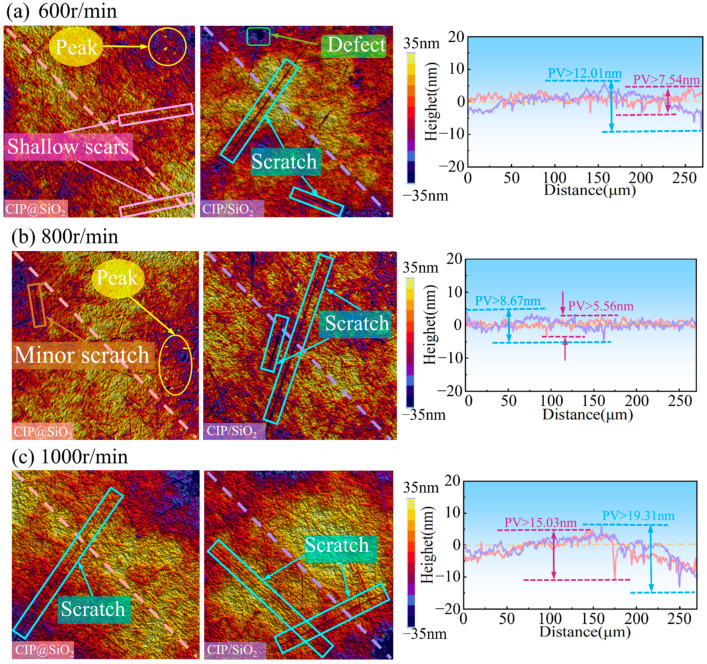
Surface topography and diagonal surface profile curves of polished fused silica with different abrasive grains at different rotational speeds.

**Figure 15 micromachines-16-00495-f015:**
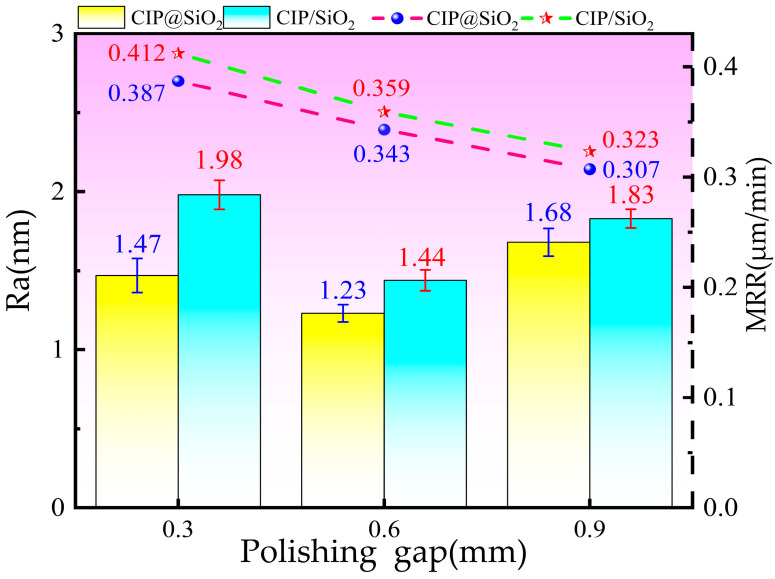
Surface roughness and material removal rate of polished fused silica with different abrasives at different gaps.

**Figure 16 micromachines-16-00495-f016:**
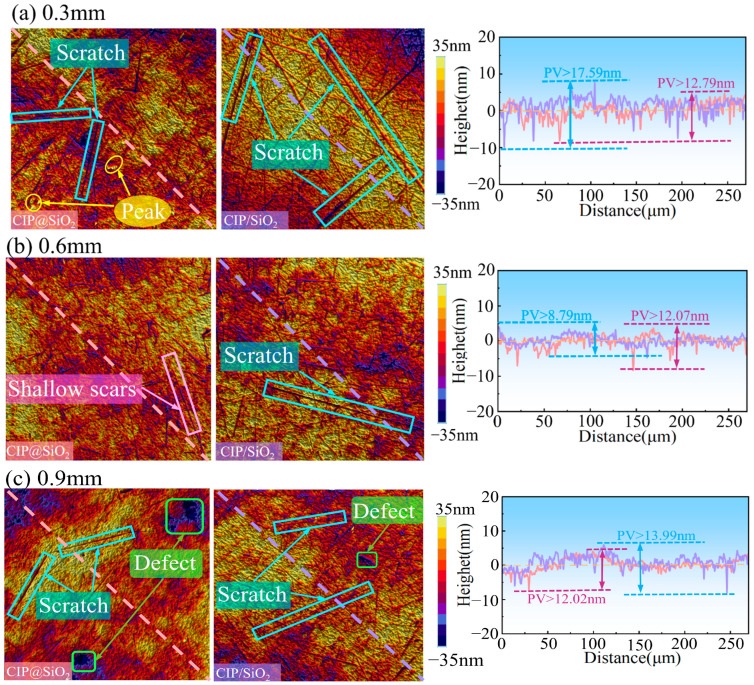
Surface topography and diagonal surface profile curves of polished samples under different abrasives and different gaps.

**Figure 17 micromachines-16-00495-f017:**
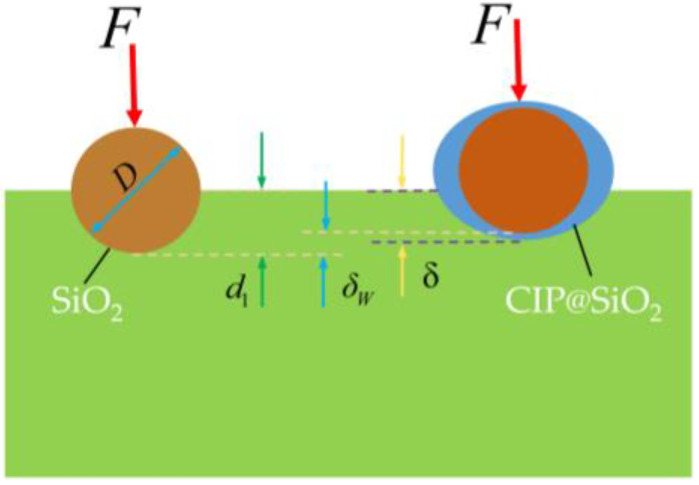
Schematic diagram of micro-contact between particles and fused silica sample during polishing.

**Figure 18 micromachines-16-00495-f018:**
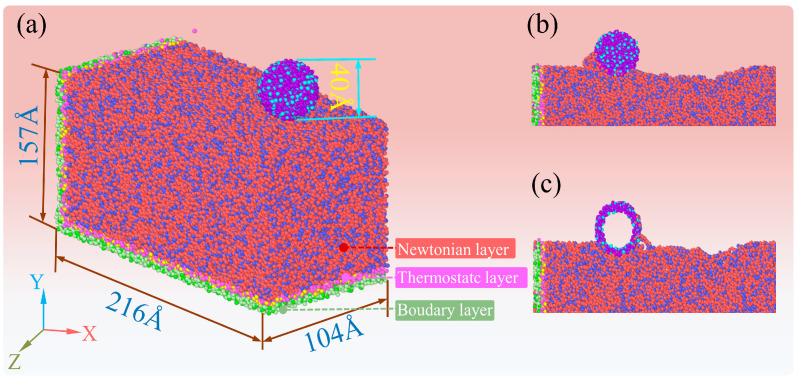
Molecular dynamics simulation model (**a**) for processing fused silica with SiO_2_ abrasive particles and SiO_2_ shells, and the corresponding intermediate cross sections SiO_2_ abrasive particles (**b**) and SiO_2_ shells (**c**).

**Figure 19 micromachines-16-00495-f019:**
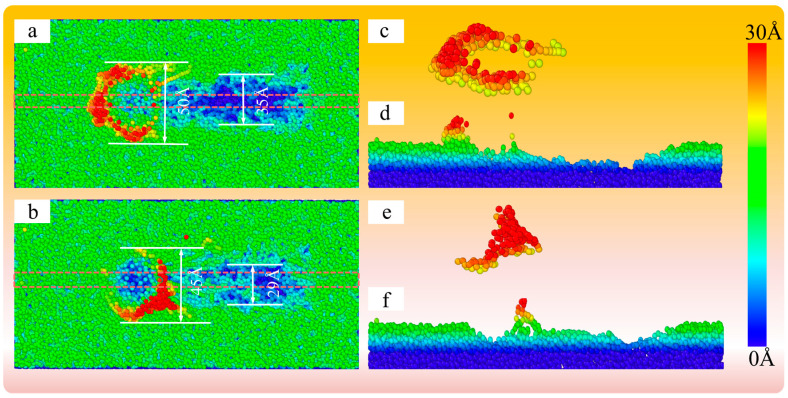
Surface topography maps (**a**,**b**) of fused silica with a processing distance of 100 Å between SiO_2_ abrasive grains and SiO_2_ shell, cross-section maps (**d**,**f**) of intermediate layer thickness, and atomic stacking maps (**c**,**e**).

**Figure 20 micromachines-16-00495-f020:**
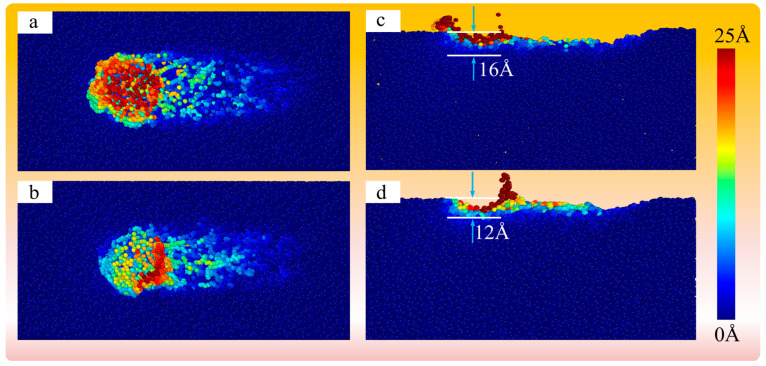
Subsurface atomic displacements diagram (**a**,**b**) and cross section diagram of interlayer (**c**,**d**) of fused silica processed by SiO_2_ abrasive particles and SiO_2_ cores.

**Table 1 micromachines-16-00495-t001:** Experimental parameters.

Factors	Level
Abrasive type	35 vol% CIP@SiO_2_ composite abrasive, 35 vol% CIP/SiO_2_ mixed abrasive (CIP 24 vol%, SiO_2_ 11 vol%)
Polishing time (min)	30, 60, 90, 120
Polishing speed (r/min)	600, 800, 1000
Polishing gap (mm)	0.3, 0.6, 0.9

## Data Availability

The original contributions presented in this study are included in the article. Further inquiries can be directed to the corresponding author.

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
