# Peer review of "Mechanism-Oriented Analysis of Core–Shell Structured CIP@SiO2 Magnetic Abrasives for Precision-Enhanced Magnetorheological Polishing"

_micromachines, 2025, doi:10.3390/mi16050495_

Round 1
Reviewer 1 Report
Comments and Suggestions for Authors
The Authors in the study titled "Mechanism-Oriented Analysis of Core-Shell Structured CIP@SiO2 Magnetic Abrasives for Precision-Enhanced Magnetorheological Polishing" investigated the impact of the placement of the abrasive system (separately in MR slurry or as a surface layer on magnetic microparticles) on the quality of the polished surface, considering variables such as polishing time, distance of the permanent magnet from the workpiece, and polishing speed. It was demonstrated that although the standard system with freely dispersed abrasive shows a more significant material removal in certain settings, the quality and lower subsurface defects are achieved using magnetic microparticles with a surface layer of abrasive. This fact was further demonstrated by the Authors using molecular dynamics simulations. The study is interesting for the field of precision surface finishing of materials and can be published in the journal Micromachines, but only after addressing the following queries:
- The Introduction section is a fairly good entry into the addressed issue, but other possibilities for increasing the long-term stability of MR slurries than using core-shell structured particles should also be given more attention, such as DOI 10.1016/j.colsurfa.2024.134444 or DOI 10.3390/ijms232012187.
- Carbonyl iron particles should not be described by the chemical formula of this originally liquid precursor, whose thermal decomposition yields the high iron content microparticles.
- The chemical (?) attachment of SiO2 to CIP should be described or schematically presented. In the current text, it is not clear how the SiO2 surface layer is attached to the magnetic microparticles.
- The use of organosilanes in the field of magnetorheological materials should be discussed more to demonstrate their effectiveness (DOI 10.1039/c4ra11842a or DOI 10.1016/j.colsurfa.2021.126871).
- To which element does the prominent peak around 2.00000 keV in Figure 3a belong?
- To more credibly demonstrate the SiO2 surface layer on CIP, a TEM image of uncoated CIP at the same magnification as the core-shell particles in Figure 5 should be used.
- The caption of Figure 7 should include the MR polishing parameters under which the sample was processed.
- What was the concentration of SiO2 abrasive freely present in the comparative MR slurry and on what basis was this concentration chosen? Is the amount of free and particle-bound SiO2 comparable?
- The intensity of the magnetic field corresponding to the stated polishing gaps should be expressed at least in Table 1.
- Why did the Authors use only the Ra parameter to describe surface roughness and not also evaluate using Rv and Rp?
- How do the Authors explain the different Ra trends over polishing time in Figure 10b (better polishing with free SiO2 distribution in MR slurry at shorter times and better polishing with SiO2 bound on CIP surface at longer times)?
- In the text, inaccuracies in the notation of indices in chemical formulas should be corrected and the notations vol% vs. v% or VSM vs. VES should be unified.
Author Response
Comments 1: The Introduction section is a fairly good entry into the addressed issue, but other possibilities for increasing the long-term stability of MR slurries than using core-shell structured particles should also be given more attention, such as DOI 10.1016/j.colsurfa.2024.134444 or DOI 10.3390/ijms232012187.
Response 1: Thank you for pointing this out. We agree with this comment. Therefore, we have enriched the discussion by incorporating additional methods for improving MR slurry stability as suggested. The modifications can be found in: Page 2, Paragraph 2, Lines 68-78.
Comments 2: Carbonyl iron particles should not be described by the chemical formula of this originally liquid precursor, whose thermal decomposition yields the high iron content microparticles.
Response 2: We sincerely appreciate the reviewer's insightful comment regarding the proper description of carbonyl iron particles. In response to this valuable suggestion, we have carefully revised the text on page 3, paragraph 2, line132 to more accurately describe the material. The modified text now states: "In the preparation of CIP@SiO2â‚‚ magnetic composite particles, high-purity carbonyl iron powder (CIP, Jiangsu Tianyi Ultra-fine Metal Powder Co., Ltd.), obtained through thermal decomposition of Fe(CO)5, was used as the core material."
Comments 3: The chemical (?) attachment of SiO2 to CIP should be described or schematically presented. In the current text, it is not clear how the SiO2 surface layer is attached to the magnetic microparticles.
Response 3: Thank you for pointing this out. We agree with this comment. Therefore, we have added a schematic diagram of the chemical connection process between SiOâ‚‚ and CIP (carbonyl iron powder) to clarify the binding mechanism. This change can be found in the revised manuscript on page 4, line 148. Additionally, the chemical connection process is described in detail on page 4, paragraphs 1, lines 142–147.
Comments 4: The use of organosilanes in the field of magnetorheological materials should be discussed more to demonstrate their effectiveness (DOI 10.1039/c4ra11842a or DOI 10.1016/j.colsurfa.2021.126871).
Response 4: Thank you for your valuable suggestions. We fully agree with this opinion. Therefore, we have provided a supplementary discussion on the use of organosilanes in magnetorheological materials in the first paragraph on page 3 ,paragraphs 1, lines 99–115 and cited key documents (DOI: 10.1039/c4ra 11842a and DOI: 10.1016/j.colsurfa.2021.126871) to demonstrate its effectiveness.
Comments 5: To which element does the prominent peak around 2.00000 keV in Figure 3a belong?
Response 5: Thank you for your comment. The peak at 2.12 keV in the EDS spectrum (Figure 3a) is definitively attributed to the Au Mα emission line (standard energy: 2.120 keV), as our samples were gold-coated (Au) prior to SEM/EDS analysis. This assignment is consistent with the characteristic X-ray lines of Au and confirms the absence of other elemental contributions in this energy range.
Comments 6: To more credibly demonstrate the SiO2 surface layer on CIP, a TEM image of uncoated CIP at the same magnification as the core-shell particles in Figure 5 should be used.
Response 6: We sincerely appreciate your valuable suggestions. We have carefully addressed your comments by making the following improvements to our manuscript:
1.Image supplementation:Added new TEM images of uncoated CIP particles (Fig. 6)
Maintained identical magnification and imaging conditions as in Fig. 6 The images clearly show the carbon shell characteristics of pristine CIP;2.Figure and text modifications:Revised the title of Fig. 6 (Page 6, Line 201)Added detailed comparative analysis in Results and Discussion (Page 6, Lines 192-199).
Comments 7: The caption of Figure 7 should include the MR polishing parameters under which the sample was processed.
Response 7: Thank you for your comment. We agree with this suggestion and have revised the text on Page 8, Line 234-235 of the manuscript to include the complete MR polishing parameters." Sample polished with 35% volume CIP@SiO2 Comparison chart of magnetorheological fluid before and after polishing at a speed of 800r/min for 2 hours in a 0.6mm polishing gap."
Comments 8: What was the concentration of SiO2 abrasive freely present in the comparative MR slurry and on what basis was this concentration chosen? Is the amount of free and particle-bound SiO2 comparable?
Response 8: We appreciate the reviewer's important question, which provided us an opportunity to better clarify the theoretical foundation of our concentration design.The free SiOâ‚‚ concentration of 11% used in this study was determined through geometric calculations (using spherical volume formula) of the core-shell structure consisting of 300nm CIP cores and 20nm SiOâ‚‚ shells. When the content of CIP@SiOâ‚‚ composite particles is 35 vol%, the SiOâ‚‚ shell naturally accounts for 11 vol%. Therefore, the comparative experiments employed the same concentration (11 vol%) of free SiOâ‚‚ to maintain consistent total abrasive content. We have provided additional explanations on page 8, paragraph 1, lines 239 to 253.
Comments 9: The intensity of the magnetic field corresponding to the stated polishing gaps should be expressed at least in Table 1.
Response 9: Thank you for your valuable suggestions. Regarding the magnetic field strength data corresponding to the polishing gap, we need to explain that due to factors such as limited experimental conditions, non-uniformity of magnetic field distribution, and dynamic changes in the polishing area, accurate measurement of magnetic field strength under various gap conditions is difficult. This study mainly focuses on the influence of relative gap changes on the polishing effect. Although specific values cannot be provided, we use a numerical control system to strictly control the gap control accuracy and adopt a standardized magnetorheological fluid ratio to ensure the comparability of the experiment.
Comments 10: Why did the Authors use only the Ra parameter to describe surface roughness and not also evaluate using Rv and Rp?
Response 10: We thank the reviewer for their comments regarding the selection of roughness parameters. In this study, Ra (arithmetic average roughness) was selected as the primary parameter for the following reasons: The polished surface exhibits relatively uniform peak-valley distribution with minimal extreme height variations. Therefore, Ra (arithmetic average roughness) was employed to characterize the overall surface roughness, which provides sufficient comparative significance for this investigation.
Comments 11: How do the Authors explain the different Ra trends over polishing time in Figure 10b (better polishing with free SiO2 distribution in MR slurry at shorter times and better polishing with SiO2 bound on CIP surface at longer times)?
Response 11:Thank you for pointing this out. We agree with this comment. Therefore, we have supplemented the explanation on Page 10, Paragraph 2, Lines 296-304of our revised manuscript to clarify the different Ra trends observed in Figure 10b. The updated text now reads:"This is mainly because the free SiO2 abrasives have higher hardness than the amorphous SiO2 bonded on CIP surfaces and exhibit greater mobility in the polishing slurry, enabling them to quickly contact the workpiece surface and efficiently remove material to achieve better surface quality faster. The softer amorphous SiO2 abrasives bonded on CIP surfaces provide a more stable material removal process that avoids over-cutting or surface damage. During prolonged polishing, free abrasives may cause surface inhomogeneity due to excessive cutting, while the softer composite abrasives can maintain more stable polishing pressure, ultimately yielding superior final surface quality."
Comments 12:In the text, inaccuracies in the notation of indices in chemical formulas should be corrected and the notations vol% vs. v% or VSM vs. VES should be unified.
Response 12: We sincerely appreciate the reviewer’s careful reading and constructive feedback. We fully agree with the importance of consistent and accurate notation in scientific writing. In response to the comment, we have made the following revisions throughout the manuscript: The term "vol%" has been uniformly adopted to replace "v%" or "V%" for clarity and consistency; VSM" (Vibrating Sample Magnetometer) has been used consistently where applicable. Any inadvertent use of "VES" (if present) has been corrected or clarified.

Reviewer 2 Report
Comments and Suggestions for Authors
In order to improve the machining precision of magnetorheological polishing, this paper conducts a mechanism-oriented analysis of the core-shell structured CIP@SiOâ‚‚ magnetic abrasives. The research content includes the preparation of the core-shell structured CIP@SiOâ‚‚ magnetic abrasives and the polishing machining experiments. The research work has certain novelty. However, there are still the following problems that need to be improved:
- In introduction, it is recommended to add the research status of accuracy and efficiency related to the main theme, especially the significance of the need to improve the accuracy of magnetorheological polishing. As is well known, magnetorheological polishing is an ultra - precision machining method, which can obtain a nearly non-damaging machined surface. How to balance the machining efficiency when further improving the accuracy in this paper? What is the relevant research progress?
- The information such as particle size parameters and concentrations for CIP@SiO2 and CIP/SiO2 are missing in Table 1. At the same time, the physical characteristic parameters of the machining object and the abrasives are lacking.
- It is necessary to carefully check for spelling errors and grammatical mistakes, such as: Line 38, Does “flexible polishing mold” mean “flexible polishing mode”?; line 43 and 260, abrasive should be abrasives; line 218, parameter should be parameters, and etc.
- In Figure 10b, the unit of time on the horizontal axis is incorrect. It should be "min" instead of "h". It is necessary to explain why the surface roughness after polishing with CIP/SiO2 decreases more rapidly than that after polishing with CIP@SiO2 during the machining time of approximately 75 minutes.
- For the core-shell structured CIP@SiOâ‚‚ magnetic abrasives, what is the appropriate thickness of the outer shell? And how does it affect the improvement of accuracy and the material removal rate? It is recommended to add relevant descriptions.
- A simulation study of surface and subsurface damage was conducted using molecular dynamics. When polishing experiments are carried out with the two types of abrasives, what is the actual subsurface damage?
- The conclusion is merely a summary of the experimental results such as surface roughness and material removal rate. It is recommended to re-condense it, especially the conclusions at the mechanism level, so as to highlight the theme of the paper.
Author Response
Dear Editors and reviewers:
Firstly, we would like to express our gratitude to you for the timely and competent comments. According to the comments, we have made careful revisions. In addition, we apologize for the carelessness in our revision process.
All revisions are highlighted in red in the revised paper.
The following is the point-to-point response to the reviewer’s comments.
For reviewer 2
Comments 1: In introduction, it is recommended to add the research status of accuracy and efficiency related to the main theme, especially the significance of the need to improve the accuracy of magnetorheological polishing. As is well known, magnetorheological polishing is an ultra - precision machining method, which can obtain a nearly non-damaging machined surface. How to balance the machining efficiency when further improving the accuracy in this paper? What is the relevant research progress?
Response 1: Thank you for your valuable feedback on the current research status of accuracy and efficiency in magnetorheological polishing (MRP). As you mentioned, balancing machining efficiency while further improving accuracy is a key challenge in the MRP field. In response to your suggestion, we have provided a detailed explanation on page 2, lines 84-107 of the manuscript, summarizing the latest progress in improving accuracy and maintaining efficiency with core-shell composite abrasive particles.
Comments 2: The information such as particle size parameters and concentrations for CIP@SiO2 and CIP/SiO2 are missing in Table 1. At the same time, the physical characteristic parameters of the machining object and the abrasives are lacking.
Response 2: We sincerely appreciate the reviewer’s valuable feedback. In response to your comment, we have supplemented the missing information as follows:1.Particle size parameters and concentrations for CIP@SiOâ‚‚ and CIP/SiOâ‚‚ have been added to Table 1 and further discussed in Lines 239–242 (Page 8). 2. The selection rationale for these parameters has been provided in Lines 246–2543(Page 8). 3. Physical characteristics of the machining object (fused silica) have been included in Lines 215–217 (Page 7).
Comments 3: It is necessary to carefully check for spelling errors and grammatical mistakes, such as: Line 38, Does “flexible polishing mold” mean “flexible polishing mode”?; line 43 and 260, abrasive should be abrasives; line 218, parameter should be parameters, and etc.
Response 3: Thank you for carefully reviewing and pointing this out. We agree with this comment. Therefore, we have made the following corrections:
1.Line 38:Revised text (Page 1, Paragraph 1, Line 38):
"This forms a 'flexible polishing mode' upon contact with the part's surface..."
2.Lines 43 & 260:
Revised text (Page 2, Paragraph 1, Line 44):
"... magnetic field, the abrasives particles are suspended and free..."
Revised text (Page 11, Paragraph 1, Line 313):
"... mixed abrasives grains caused greater damage..."
3.Line 218: Revised text (Page 9, Paragraph 1, Line 243):
"... Specific polishing parameters settings are shown in Table 1...."
Comments 4: In Figure 10b, the unit of time on the horizontal axis is incorrect. It should be "min" instead of "h". It is necessary to explain why the surface roughness after polishing with CIP/SiO2 decreases more rapidly than that after polishing with CIP@SiO2 during the machining time of approximately 75 minutes.
Response 4: We sincerely appreciate your careful review and valuable suggestions. We have made the following revisions to the manuscript:1.Correction in Figure 11b:The time unit on the horizontal axis has been corrected from "h" (hours) to "min" (minutes).
2.We added an explanation for the faster decline in CIP/SiO2 roughness on page 10, paragraph 2 line 296-304:"This is mainly because the free SiOâ‚‚ abrasives have higher hardness than the amorphous SiOâ‚‚ bonded on CIP surfaces and exhibit greater mobility in the polishing slurry, enabling them to quickly contact the workpiece surface and efficiently remove material to achieve better surface quality faster. The softer amorphous SiOâ‚‚ abrasives bonded on CIP surfaces provide a more stable material removal process that avoids over-cutting or surface damage. During prolonged polishing, free abrasives may cause surface inhomogeneity due to excessive cutting, while the softer composite abrasives can maintain more stable polishing pressure, ultimately yielding superior final surface quality."
Comments 5: For the core-shell structured CIP@SiOâ‚‚ magnetic abrasives, what is the appropriate thickness of the outer shell? And how does it affect the improvement of accuracy and the material removal rate? It is recommended to add relevant descriptions.
Response 5: Sincere thanks to the reviewing experts for their important suggestions on optimizing the thickness of SiOâ‚‚ shell. Due to experimental limitations, we have not yet systematically studied the influence of shell thickness. The core objective of this study is to systematically compare the core-shell structure CIP@SiO2 The essential differences in surface processing quality and material removal efficiency between and traditional mixed CIP/SiOâ‚‚ abrasives.
Comments 6: A simulation study of surface and subsurface damage was conducted using molecular dynamics. When polishing experiments are carried out with the two types of abrasives, what is the actual subsurface damage?
Response 6: Thank you for raising this crucial question. Although this study mainly focuses on surface roughness and material removal rate indicators, we recognize that characterizing subsurface damage (SSD) will make the research more comprehensive. According to the molecular dynamics (MD) simulation results (Figure 19), CIP@SiO The SSD depth predicted by the â‚‚ abrasive is approximately (16 Å), which is significantly smaller than the CIP/SiOâ‚‚ mixed abrasive (approximately 12 Å), attributed to the stress localization effect of the SiOâ‚‚ shell. This result is consistent with the observed lower surface roughness (Ra) in the experiment.
Comments 7: The conclusion is merely a summary of the experimental results such as surface roughness and material removal rate. It is recommended to re-condense it, especially the conclusions at the mechanism level, so as to highlight the theme of the paper.
Response 7: Thank you for pointing this out. We agree with this comment. Therefore, we have revised the conclusions section to focus more on mechanistic-level insights and removed redundant experimental results descriptions. This change can be found in the revised manuscript on Page 18, Paragraph 2-3, Lines 520-540.
In summary, we would like to express our gratitude to the reviewers for their helpful and insightful comments again. We will be gladly to answer any other questions if the reviewers deem necessary. |

Round 2
Reviewer 1 Report
Comments and Suggestions for Authors
There are minimal inaccuracies in the manuscript (referencing Fig. 10 (a) and (b) (line 279), which actually refer to Fig. 11). However, these inaccuracies will certainly be corrected during the proofreading process. To conclude, the Authors have addressed the points that the Reviewer raised and answered all the questions, so the study can be published in Micromachines.
Reviewer 2 Report
Comments and Suggestions for Authors
In accordance with the reviewers' comments, the author has made earnest revisions and provided meticulous responses. It is recommended for publication.